# Learning to Cycle: From Training Wheels to Balance Bike

**DOI:** 10.3390/ijerph19031814

**Published:** 2022-02-05

**Authors:** Cristiana Mercê, Marco Branco, David Catela, Frederico Lopes, Rita Cordovil

**Affiliations:** 1Centro Interdisciplinar de Estudo da Performance Humana, CIPER, Faculdade do Motricidade Humana, Universidade de Lisboa, 1499-002 Cruz-Quebrada, Portugal; marcobranco@esdrm.ipsantarem.pt (M.B.); cordovil.rita@gmail.com (R.C.); 2Escola Superior de Desporto de Rio Maior, Instituto Politécnico de Santarém, 2040-413 Rio Maior, Portugal; catela@esdrm.ipsantarem.pt; 3Motor Behavior, CIEQV, Instituto Politécnico de Santarém Branch, Complexo Andaluz, 2001-904 Santarém, Portugal; 4Faculdade de Motricidade Humana, Universidade de Lisboa, 1499-002 Cruz-Quebrada, Portugal; fred.lopes3@gmail.com

**Keywords:** balance bike, bicycle with training wheels, learning to ride a bicycle, constrains, learning paths, cycling, Portugal

## Abstract

Background: Learning to cycle is an important milestone in a child’s life, so it is important to allow them to explore cycling as soon as possible. The use of a bicycle with training wheels (BTW) for learning to cycling is an old approach practiced worldwide. Most recently, a new approach using the balance bike (BB) has received increased attention, and several entities believe that this could be most efficient. Drawing on the work of Bronfenbrenner (1995) and Newel (1986), this study aimed to analyse the effect of BB’s use on the learning process of cycling independently. Methods: Data were collected in Portugal from an online structured survey between November 2019 and June 2020. Results: A total of 2005 responses were obtained for adults and children (parental response). Results revealed that when the BB’s approach was used, learning age (LA) occurred earlier (M = 4.16 ± 1.34 years) than with the BTW’s approach (M = 5.97 ± 2.16 years) (*p <* 0.001); or than when there was only the single use of the traditional bicycle (M =7.27 ± 3.74 years) (*p <* 0.001). Conclusions: Children who used the BB as the first bike had a significantly lower LA than children who did not use it (*p <* 0.001). To maximize its effects, the BB should be used in the beginning of the learning process.

## 1. Introduction

Humans have different natural modes of locomotion, such as walking and running. With the cultural evolution of our species, the bicycle was invented as a transport vehicle, being more efficient, economic and less tiresome than our natural modes of locomotion [1]. Nowadays, this invention won a very important role in human life; it is used everywhere for transportation, exercise, sports competition, or simply for recreation [2,3]. Cycling also proved to be an activity that improves health. It has a positive relationship with cardiorespiratory fitness in youths, cardiovascular fitness in adults, and a strong inverse relationship with all-cause mortality, cancer mortality and morbidity in middle-aged and elderly people [4]. In children, cycling also has several health benefits, like better cardiorespiratory fitness, less body fat, and less incidence of metabolic syndrome [5]. There are also social benefits, such as the development of relational and emotional skills, promoting fun play moments where children can interact with other people, and make new friendships [6,7]. In addition, cycling allows for a greater exploration of the environment mobility, enabling children to become more independent and active [8]. Cycle trains are a good example of this, children travel to school by bicycle and stop at their colleagues’ houses increasing the “train” until school [9]. Most recently, the active transport in children, including cycling, has also revealed an positive association with academic achievement and cognition [10]. For all these reasons, learning to ride a bicycle is an important milestone in children’s lives [11], so it is important to allow children to explore cycling as soon as possible.

The present study draws on the theoretical juxtaposition of the Bioecological Theory of Bronfenbrenner [12] and Newell’s model of constraints [13] applied to the learning pathways of bicycles sequences that children go through until they are able to cycle independently and without training wheels.

According to Bronfenbrenner, the child’s development occurs within interactions and relationships between the child and his/her environment [12,14]. The different layers of environment affect the child’s development, including motor development and the learning of new skills, such as learning how to cycle. The initial model proposed by Bronfenbrenner [14] considered the following layers: micro-, meso-, exo-, and macrosystem [14]. At a later stage [12], time was included into the model and the chronosystem dimension was added. The different microsystems consist in a set of environments where the child can engage in face-to-face interactions with other people; for example, family, friends, or community institutions like the school are examples of microsystems. If the microsystems the child interacts with value cycling, and if the child has access to a bicycle since an early age, it is more likely that he or she will learn to ride a bicycle earlier than if cycling and having a bicycle are not valued or prioritized. Parental encouragement is a key factor not only to cycle learning [15], but also for increasing cycle practice [16]. The mesosystem comprises the interactions between the different microsystems, for example, the relationship between the child’s family and the school. If the school launches a “bike to school” campaign and the family has a good relationship and an active participation in the school, it is more likely that they will join that campaign [16]. The exosystem includes contexts where the child is not directly involved but that can have an indirect effect on him/her, such as the availability of community programs for cycling in the child’s neighbourhood, or the media promotion of active transport and cycling. The existence of a community program to promote cycling in a family can enhance bicycle use and learning from an early age [17]. The macrosystem consists of societal, cultural and global influence, which can include the cultural value given to cycling, the role attributed to gender, or simply the laws and governmental policies. If the government promotes safe conditions for cycling, for example through bike paths’ construction or protective laws for cyclists, an increase in cycling is expected [18]. Finally, the chronosystem adds the dimension of time; for example, the era in which the child lives also influences the value given to cycling, the age at which the child’s parents will give him/her a bike, and the type of bike the child will be given (if any). 

In our perspective, when looking at the milestone of learning to ride a bike, Bronfenbrenner’s theory shares a common ground with Newell’s model of constraints [13], namely in terms of what Bronfenbrenner and Morris [19] describe as four fundamental properties (person, context, time and process), which dynamically interact with each other in order for developmental acquisitions to occur. The process is the central intermediate element of the model as it represents particular forms of interaction that occur over time between the person and the environment. These reciprocal interactions, designated of proximal processes, progressively become more complex and are considered the key agents of human development [19]. However, the degree of influence these proximal processes have on development varies according to the interrelationship given by the evolving person’s characteristics, the immediate and more distal environmental contexts, and the time periods of these interactions [20]. Similarly, in a more microscopic scale, according to Newell, movement arises from the dynamic interaction between individual, task and environmental constraints [12,13,14]. Individual constraints consist of the features of the system itself, like age or motor competence. Probably, children with a better motor competence and a greater motor repertoire will learn to ride a bicycle more easily [21]. Task constraints consist of features related to the task itself that can be modified, such as the instrument used, its duration and its frequency. For example, several institutions believe that the balance bike can be more efficient for learning than the bicycle with training wheels [22,23]. Finally, environmental constraints are features related to the physical environment like the weather, or to the sociocultural factors like the family context. In this sense, the dynamic proximal processes between the child and the environment advocated by Bronfenbrenner’s theory are also present in Newell’s model. According to this model, these proximal interactions between the different constraints are fundamental, and if any constraint changes, the resultant movement changes. Sometimes constraints change mildly (e.g., when the individual constraint of the height of the child changes it might be necessary to adjust the height of the bike), but sometimes constraints change more abruptly (e.g., changing a task constraint such as taking the training wheels out will interact with the child’s ability to keep balance). 

To learn how to ride a bicycle, the combination of constraints and possible pathways are endless. For example, the child can learn alone, with parents, friends; can practice in the street, cycle path or dirt; use a balance bike, bicycle with training wheels, or simply the traditional bike.

The learning process is always individual and complex. Each system, each human being, is unique and is influenced by the sociocultural environment and by different constraints [12,13,14]. The variability of possible pathways to learn how to cycle is probably one of the reasons why the better or the most efficient methodology and type of bike used for learning is still not consensual.

The use of the bicycle with lateral training wheels (BTW) is a worldwide practice, however not everyone agrees with this approach [24,25]. Recently, the use of the balance bike is increasing; in Portugal, one of the biggest sporting goods retailers started selling this bike in 2012–2013, and some of the biggest supermarkets also started in this decade, which may also have contributed to making BB more accessible and popular. A balance bike (BB) consists of a bicycle without training wheels or pedals, so children should use their feet against the ground to propel themselves. Several institutions, including the Portuguese Cycling Federation (PCF) and the Biciculture House in Portugal, believe that using a BB instead of the traditional BTW improves the learning process. For this reason, some initiatives of the PCF, such as the “Cycling for Everyone” and the “Cycling Goes to School”, provide balance bikes for children who do not know how to ride [22,23]. 

While the traditional and old approach with BTW allows children to explore the pedalling being balanced by the training wheels, the new approach with BB works the other way around, allowing children to first explore the balance in the bicycle, and then introducing the pedalling (Figure 1). Despite the empirical experience of bicycle instructors that prefer to use BB and the positioning of recognized entities like PCF, the scientific literature that supports balance bike’s use is very scarce. In this sense, the present article aimed to study the influence of balance bike’s use on the process of learning to ride a bicycle independently, adopting a bioecological approach to such a relevant acquisition in terms of children’s motor development. More specifically, we aimed to: (i) verify if the BB’s use is related to a possible decrease in the learning age of independently cycling (LA) over decades; (ii) identify the most common learning pathways of a bicycles sequence (learning paths); (iii) verify if the learning paths are related with the LA; and (iv) analyse and compare the LA between children who used and did not use BB.

## 2. Materials and Methods

### 2.1. Survey

The data collection was carried out within the scope of the Learning to Cycle project (L2Cycle), which developed a retrospective online survey to access the cycle LA [26]. This retrospective method has been used before to collect the LA of several other milestones, e.g., roll over, sit up, stand alone, walk, first words, smiling or crawling [27,28]. To create the L2Cycle survey, several phases have been completed; during the pilot phase, an initial version of the survey was developed by a group of four experts in child development and was tested online on 485 participants. A sub-sample of 30 participants was additionally inquired about the comprehension of the survey. After that, some adjustments were made. For example, one group related to the dates of acquisition of different motor milestones was deleted, and some questions were reformulated to improve clarity according to the respondent’s suggestions. At a second stage, the survey was discussed with a group of five international experts who provided further suggestions (e.g., adding questions regarding mother tongue and different seasons of the year). Finally, the survey was translated for different languages and is now available in 10 languages (Portuguese—from Portugal and Brazil, English, German, Croatian, Finish, French, Dutch, Italian, Japanese and Spanish). For the current article, only the Portuguese data were analysed. The final Portuguese version was launched online on 22 November 2019 and data for the current study were collected between that date and 8 June 2020. The survey was publicized in the national conference on Child Development and disseminated through social media (Facebook, Instagram, WhatsApp), and by email. In addition, partnerships with the PCF and children’s and parent’s magazines were established for dissemination on their websites and paper magazines.

The survey takes approximately five to fifteen minutes to complete (depending on the number of children), it is anonymous, and is comprised of three sections:“About you”—Questions about the participant’s own experience and biographical data (e.g., place of residence, age, gender, physical activity habits, if they know to ride a bike, if not—why not, if yes—when did they learn, what types of bikes were used and in what sequence, where did they learn, who taught them, how often do they ride a bike, what do they use it for).“About your older child” (to be completed only if the participant has children)—These questions are the same as the questions in the first group but regarding the participant’s older child.“About your younger child” (to be completed only if the participant has more than one child)—These questions are the same as the questions in the first group but regarding the participant’s younger child.

This survey was approved by the Ethics Committee of the Faculty of Human Kinetics (approval number: 22/2019).

### 2.2. Sample

The survey was completed with information regarding 2386 participants. For the present study, only participants born during or after the decade of 1960–69 and who could ride a bicycle independently were considered (n = 2005). Participant’s age ranged from 2.39 to 60.18 years (M = 27.97 ± 14.7 years). In order to analyse differences in learning to ride a bike across generations, the birth decades of the participants were considered. Regarding geographical location, we collected data from participants in all 20 Portuguese districts and the two autonomous regions, Madeira and Azores. Descriptive data of the sample is presented in Table 1.

### 2.3. Statistical Analysis

Data extracted from LimeSurvey was organized and codified by a Matlab routine specifically developed for this purpose. The data were later processed in the software Statistical Package for the Social Sciences (SPSS, version 25). 

Analyses of frequency and chi-square tests were used to investigate the differences in the percentage of BB’s use between consecutive decades. One-way ANOVAs were performed to assess differences in the LA across decades and between different learning paths (i.e., considering the order of use of the BB). In cases of non-homogeneity, the Welch correction was applied. To investigate significant differences between groups, the Bonferroni or the Games Howell post-hocs were used, depending on the existence or not of homogeneity of variances [29]. The level of significance was set at 0.05. 

### 2.4. Sample Calculation

Sample calculation was performed a posteriori with the software G*Power (version 3.1.9.7.). For this calculation, it considered the effect size of the main variable, age learned, from the data of test’s version, which revealed an effect size of 0.1. A one-way ANOVA was performed on the calculation, which considered the question with the lowest sample, 1341, and the higher number of groups, 8, with a significance level of 0.05. This sample calculation estimated an observed power of 0.76.

## 3. Results

### 3.1. Learning Age over Decades

Learning age changed significantly over the decades (F(7, 786) = 41.79, *p* < 0.001, ηp2 = 0.07). Considering consecutive decades, only a non-significant increase between 1960–69 and 1970–79 (Figure 2) was found. After that, the LA always decreased, with significant differences between 1970–79 and 1980–89 (*p* = 0.01), and between 2000–09 and 2010–2019 (*p <* 0.001).

### 3.2. Use of BB and BTW over Decades

Results regarding the types of bicycles used to learn indicate that the percentage of people using the BB has increased over time from 9.6% (for people born in the 1960′s) to 49.2% (for people born between 2010 and 2019). The percentage of people using the BB increased rapidly in this millennium, since when analysing consecutive decades, we found significant differences between the decades of 1990–99 and 2000–09 (χ2(1) = 6.32, *p =* 0.012); and between 2000–09 and 2010–2019 (χ2(1) = 55.02, *p <* 0.001) (see Figure 3).

The percentage of people using the bicycle with two training wheels (BTW) has significantly increased over several decades, more specifically between 1960–69 and 1970–79 (χ2(1) = 17.62, *p <* 0.001), between 1970–79 and 1980–89 (χ2(1) = 11.34, *p <* 0.001), and between 1980–89 and 1990–99 (χ2(1) = 19.90, *p <* 0.001). This use stabilised around the percentage of 85% between 1990–99 and 2000–09, and having significantly decreased for the first time between 2000–09 and 2009–2019 (χ2(1) = 10.78, *p =* 0.001), reached the value of 75.2%.

Lastly, the percentage of people using the bicycle with one training wheel (B1TW) remained relatively stable between the decades of 1960–69 and 1980–89. It only increased once between the 1980–89 and 1990–99 (χ2(1) = 10.31, *p =* 0.001), and then dropped twice consecutively between 1999–00 and 2000–2009 (χ2(1) = 4.80, *p =* 0.028), and 2000–2009 and 2010–2019 (χ2(1) = 30.04, *p <* 0.001).

### 3.3. Learning Paths

The type of bikes and order in which those bikes were used during the learning process defines the different learning paths that were used. We considered the possible use of four types of bikes during the learning process: the balance bike (BB), a bike with two training wheels (BTW), a bike with just one training wheel (B1TW), and the traditional bike with no training wheels (TB). The learning paths emerge from any combination between the order of use of these bikes that ends with the TB. In the present article, the learning paths are represented by a sequence of four numbers, the position of the number represents the type of bike used and its value represents the order. More specifically, the first digit represents the BB, the second represents the BTW, the third represents B1TW, and the fourth represents the TB. So, if the child presented a learning path of 1002 it means that the BB was used in first place, the BTW or B1TW were not used, and the TB was used in second place. If one digit is repeated (e.g., 1102), it means that the child used those bikes simultaneously.

Of all the possible combinations, we found 29 different learning paths in our sample, but only the learning paths that were used by at least 30 participants were considered for analysis (Figure 4).

Results indicated that the LA is significantly different depending on the learning paths used (F(7, 194) = 26.83, *p* < 0.001, ηp2 = 0.08). Descriptive statistics of the LA according to the different learning path and results of the post-hoc analyses are presented in Table 2.

The learning path with the lowest LA (M = 4.16 ± 1.34 years) was the one where the BB was used first, and then TB (1002). Considering these values, and using the mean minus the SD as a reference, we believe that by two and a half years of age, children seem to be ready to start using the balance bike. People who used the BB first and then TB had a significantly lower LA (*p <* 0.001) than people who used any of the others learning paths, except using the BB first, two training wheels second, and then TB (1203). The traditional learning approach, which starts by using the two training wheels and then TB (0102) had a mean LA of 5.97 ± 2.16 years. The learning path with the highest LA was the single use of the TB (0001), with a mean age of 7.27 ± 3.74 years, a value significantly higher than all the other learning paths (*p <* 0.001).

The percentage of use of each learning path over the decades is shown in Figure 5, and it is possible to verify that the percentage of the learning paths with lower LA, as 1002 and 1203, increases; while the one with higher LA, 0001, decreases.

### 3.4. Order of Use of the Balance Bike

Considering not the learning path, but the order of use of the balance bike in the learning process, there were significant differences in LA depending on the moment the BB was used (F(1, 4) = 9.88, *p* ≤ 0.001, ηp2 = 0.02). The lowest LA occurs when the BB is used first (M = 5.13 ± 2.89 years), while the highest LA occurs when the BB is not used (M = 6.32 ± 2.13 years). The group who used the BB first learned at a significantly earlier age than the groups that never used it (*p <* 0.001) or that used it in 4th place (*p <* 0.001). 

## 4. Discussion

### 4.1. Relation between BB’s Percentage of Use and the LA over Time

Although the BB’s boom in Portugal was recent, our results indicated that at least since the 1960s some people mentioned using it in the process of learning to cycle independently. Looking from a historical perspective, the BB is very similar to the first bicycle model. The bicycle was created in 1817 by Karl Drais, and it consisted of a wooden prototype just with two wheels, without chain, brakes or pedals. Therefore, riders should propel the bike by pushing the floor with their feet [1,30]. Maybe this bicycle model persisted in some way over time. It is also possible that even after the general commercialization of the training wheels, some people still chose to remove the pedalboard and let children play with the bicycle instead of using the training wheels. We could identify the biggest boom in the use of the BB between 2000–09 and 2010–2019, which coincides with the decreases in the use of B1TW in the decade of 2000–09, and of BTW and B1TW in the decade of 2009–2019. One of biggest sport articles retailers in Portugal started to sell BBs in 2012–2013, and some of the biggest supermarkets also started to commercialize it around the same time. In this decade, the media also started to include images of balance bikes in commercials. The bigger dissemination of the BB also lead entities like PCF to include it in their cycling programs [22,23]. Some municipalities have even started to make BBs available in preschools to promote earlier cycling. All of these interactions between the macrosystem (cultural views in biking and healthy lifestyles), exosystem (cycling programs in the municipality and BB incorporated in media), mesosystem and microsystem (opportunity to explore the BB in school and with friends), added to the fact that the BB became more accessible to consumer, contributed to the significant increase of the BB’s use.

Children born during the last decade (i.e., 2010–2019) had the lowest LA compared to the other groups. However, the results of this decade should be considered with caution. Due to the historical proximity of this period, some of the participants born in the last years of this decade still have not learned how to ride a bicycle. Thus, early learners might be slightly over-represented in the last decade. Nevertheless, the tendency for a significant decrease in LA across decades was clear. Considering that the BB’s use increased significantly in the last two decades, it is possible that one of the factors associated with the decrease in LA is the increase in the use of BB. 

### 4.2. Learning Paths

We found a great variability of learning paths in our study, which underlines the fact that the same motor developmental state can be achieved over different pathways [31].

The most frequent learning path was the traditional approach (n = 630), using first the bicycle with two training wheels and then the traditional bike (learning path 0102). This data reinforces the idea that training wheels are a practice ingrained in the culture of learning to ride a bicycle. The second most frequent learning path is the one with the higher LA, the single use of TB (n = 404, learning path 0001). Although it does not seem to be a path that facilitates learning, this high frequency might result from a lack of availability of other type of bike. If the child has no opportunity to explore the BB or the training wheels, he/she probably will learn by just using the TB. It is interesting to note that the use of this pattern decreases over the decades (Figure 5), possibly due to the greater accessibility of training bikes such as the BB or the BTW. The first use of two training wheels followed by one training wheel and then the TB (learning path 0123), follows as the third most frequent path (n = 364), highlighting once more the training wheels culture. After this, using first the BB and then the TB is the next most frequent learning path (n = 54, learning path 1002). The BB’s use has significantly increased (*p <* 0.001) in the last decade (Figure 2), so it is expected that the frequency of the learning paths involving the BB, and particularly this new approach for learning, will increase in future.

The child’s learning path occurs in specific socioecological contexts, from proximal to distal [12], and is shaped at every moment by the interaction between the existent constraints [13]. The parents support during cycling learning is related to the microsystem layer; the community culture and cycling programs, to the mesosystem layer; the media promotion of training wheels or balance bike, to the exosystem layer; and a culture that values and promotes cycling, to the macrosystem layer. All these environments have the potential to shape the child’s learning path. In addition, the child’s individual constrains will also influence the learning process. For example, a poor body composition (BC) is associated with a lower balance ability [32,33,34,35]. Considering that balance is fundamental for cycling, and particularly challenging in the initial stages of learning, children with a poor BC will probably have more difficulty in learning how to cycle independently. The lack of balance can also interact with other individual constraints, such as the child’s motivation to learn. If a child constantly struggles to keep balance and falls frequently during the first stages of learning, he/she will be more likely to develop a fear of falling and to start avoiding cycling to prevent injuries. Conversely, if the child has a good motor competence, it is expected that he/she experiences more success during learning, feels more motivated, and learns to ride a bicycle earlier [36,37]. Finally, the task constraints also play an important role in the learning process. The fact that different learning paths, using different types of bikes, significantly correlated with LA in our study highlights the importance of the task constraints in the dynamic process of learning how to ride a bike.

The most successful path for learning (i.e., the path with the lowest LA, around four years of age) seems to be to use the BB first and then the TB (1002). On the other hand, using the two training wheels first and then TB (0102) seems to postpone learning to a later age (around six years of age in our study). According to our data, and not considering other potential confounding variables, by directly comparing these two approaches (Figure 1), it seems that the newest approach with the balance bike promotes a faster learning than the older, with training wheels. In average, in the present study, children who transitioned directly from the BB to the TB learned to ride 1.81 years earlier than children who transitioned from the TW to the TB. However, considering the weaknesses inherent to the methodology of a retrospective survey, and the fact that the sample is not distributed equally by genres and decades, these conclusions must be analyzed with caution.

By analysing the learning paths sorted increasingly by LA (Figure 4), it is possible to verify that in the first three paths, with the lowest LA, the BB was used the first. In the fourth path, the BB was used third, and in the last four paths, with the highest LA, it was not used. This ordering of patterns seems to confirm, only for the data presented, the association between the use of BB in the learning process and the lowest LA. Some authors consider that balance is the most difficult challenge in the process of learning how to cycle [24,25]. The balance bike improves balance from an early stage, not focusing on the pedalling coordination, and maybe this is the key for its success.

The BB allows children to explore several movement patterns while using it; they can walk, run, propel the bike with both feet or just one, and can also explore the flight phase when they experience balance for increasing amounts of time without any contact of the feet with the ground. While doing this, children are exploring and learning to control their centre of gravity and the bicycle’s centre of gravity, as they learn to keep balance on the bicycle. 

With the BTW, children develop first the ability to pedal, and balance is not a challenge because it is guaranteed by the training wheels. Therefore, when children transition from the BTW to the TB removing the training wheels, they have to learn how to balance and there is a greater instability associated with the pedalling. This approach seems to pose a greater challenge than mastering balance first and feet coordination afterwards. It should be noted that all the paths fulfil the purpose—all allow children to learn to ride a bicycle—but some of them are faster than others. 

The learning path with the highest LA consisted of the single use of TB, with a mean of 7.27 ± 3.74 years. In this approach, the initial challenges are great since there are no training wheels to guarantee the balance and the pedals are already there to be used. The child should simultaneously learn how to balance, pedal, break and turn. This seems to be a too much complex task, leading to a longer duration of the learning process. 

### 4.3. Order of Use of Balance Bike 

The importance of using the BB at the beginning of the learning process is clear if we look at the LA according to the order of use of the balance bike learning path (Figure 6). Using the BB first afforded a significantly lower LA than not using it (*p <* 0.001). The task constraint of using the BB influences the learning process [13], but that influence should occur earlier in the learning process, since as BB ceases to be prioritized, its effect decreases. Possibly, this happens not because of the BB itself, but because of the introduction of other types of bikes that require different types of adaptations from the child and cause more noise in the learning process. When the BB was not the first bike used, it generally means that children started to explore the pedalling before testing their balance, and exploring balance at a later stage does not seems to be the best option since it costs time. When BB is used in the last place, the effect is almost lost and the LA differs significantly from when it is used in the first place (*p =* 0.022). 

### 4.4. Strengths and Weaknesses

The major strength of this study was to address the existent gap in the literature concerning the influence of using the BB on this learning process. Although our results clearly support the general feeling that exists among bike instructors that the BB accelerates learning, due to the characteristics of this study (i.e., online survey), we could not analyse the learning process in a more individual basis. The results show that learning how to cycle independently is a process quite sensitive to the task constrains, specifically to the type of bicycle used, but the influence of specific individual constraints, such as body composition [32,33,34,35] or motor competence [21] on this task should be addressed in studies with a different design (e.g., smaller sample of children followed longitudinally during the learning process). This type of study would also allow us to better understand the process of mastering to control the balance bike and to explore its flight phase during the initial learning stages. Finally, the comparison between the learning process among different cultures and genders can be explored in the future.

The main weakness of the study is inherent to its typology; as this study was a retrospective survey, the recall risk is possible, i.e., the participants could not remember accurately the details asked [38,39]. Considering that the recall risk may be higher in older participants, as a strategy to control and minimize this possible bias, the responses of participants born before the decade of 1960–69 were not considered. In addition, the younger and the older participants could interpret the questions differently; as strategies to avoid this, the questions were developed and discussed in order to be simple, clear and objective. Prior to the survey’s application, 30 participants aged between 18 and 60 were asked about the comprehension of the survey. Other questions, such as the age of first approach to cycling or the cycling frequency during learning were not included. We believe these are important questions, but according to the feedback of the interviews, they would be difficult to address in a retrospective survey study. Finally, another limitation is related with the sample; although it is not small (n = 2005), the number of males and females by each decade, and the number of participants between decades, are not equivalent. The sample includes more females than males, especially in the older decades, which are possible limiting factors of the results.

## 5. Conclusions

To our knowledge, this is the first large scale study to investigate the influence of using a BB in the process of learning to ride a bicycle independently. Our results indicate that using a BB, particularly during the first stages of the learning process, leads to a significant decrease in the LA for this motor milestone. However, considering the study’s design and its weakness, the extrapolation of these results should be considered with caution. The use of the BB has been increasing throughout the decades, accompanied by a decrease in the average age for learning, which in Portugal has been more marked since the beginning of the millennium. There are different benefits of learning how to cycle earlier. For example, children who begun to cycle at an early age are more likely to have a healthy weight in the subsequent school years [40], they can have fun moments cycling outdoors with peers or family, they develop motor components, and mature their social and emotional skills [6,7]. Although a great number of learning paths will always continue to exist, it seems that the sooner children master balance, the earlier they will be able to control the TB. For the present data, the difference in the LA for cycling independently varied by two to three years depending on the learning path and the type of bikes used. This temporal gap could have an impact in a child’s life, so it is important to promote the best approach for learning how to cycle as soon as possible, which seems to be the one that uses the BB first. Based on the data, it is suggested to start learning to cycle at about two and half years of age by using the balance bike.

## Figures and Tables

**Figure 1 ijerph-19-01814-f001:**
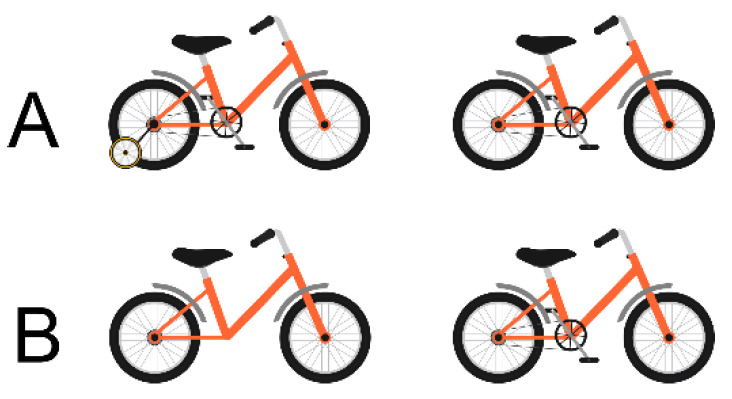
Old and new approaches for learning to cycle independently: (**A**) using training wheels; (**B**) using the balance bike (BB).

**Figure 2 ijerph-19-01814-f002:**
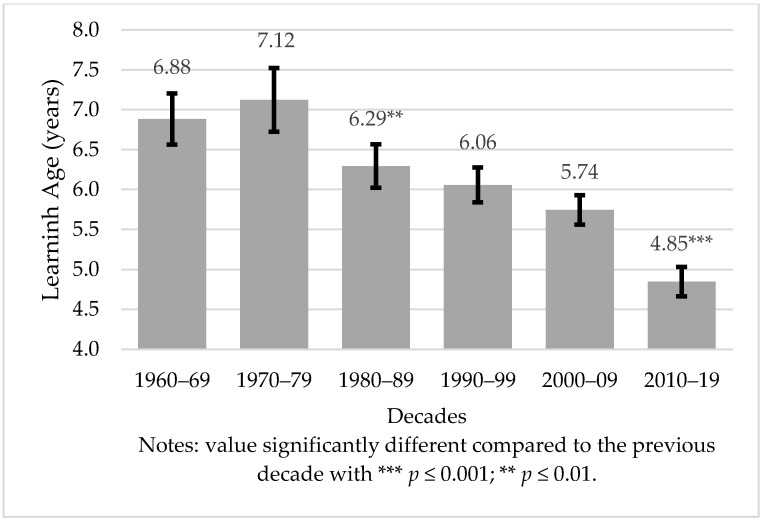
Evolution of learning age according to decades; mean and 95% confidence interval.

**Figure 3 ijerph-19-01814-f003:**
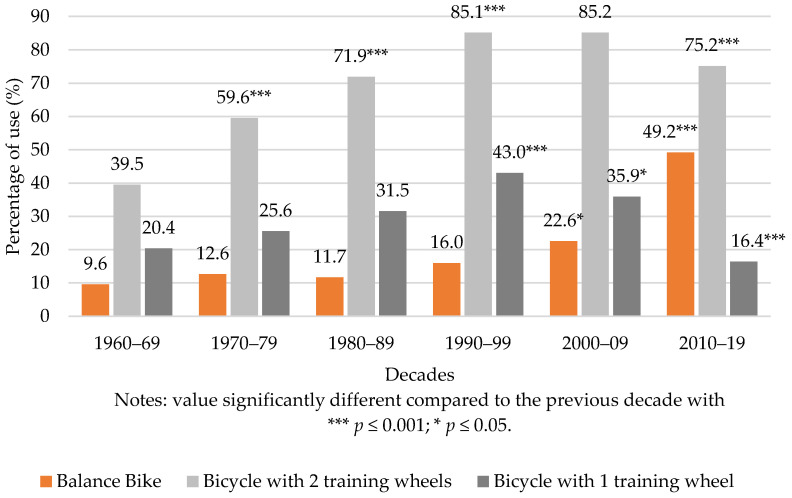
Percentage of use of BB, BTW and B1TW according to decades.

**Figure 4 ijerph-19-01814-f004:**
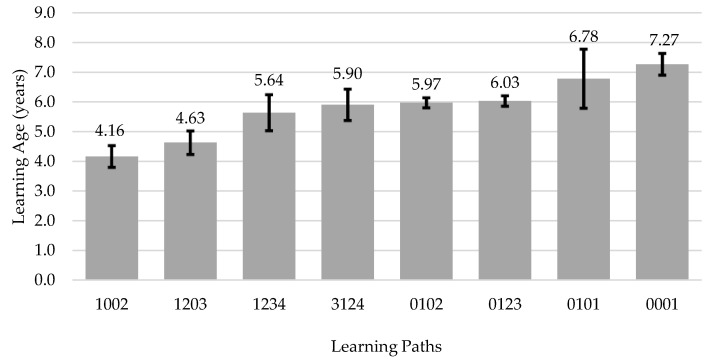
Learning age according to learnings paths; mean and 95% confidence interval. Notes: first digit in learning path—balance bike; second—bicycle with 2 training wheels; third—bicycle with 1 training wheel; fourth—traditional bicycle.

**Figure 5 ijerph-19-01814-f005:**
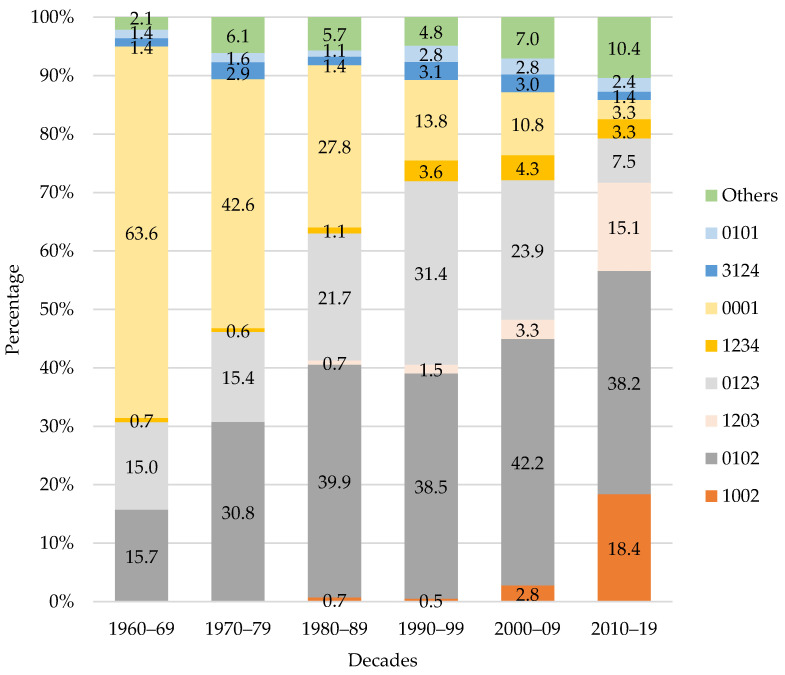
Percentage of learning paths by decade.

**Figure 6 ijerph-19-01814-f006:**
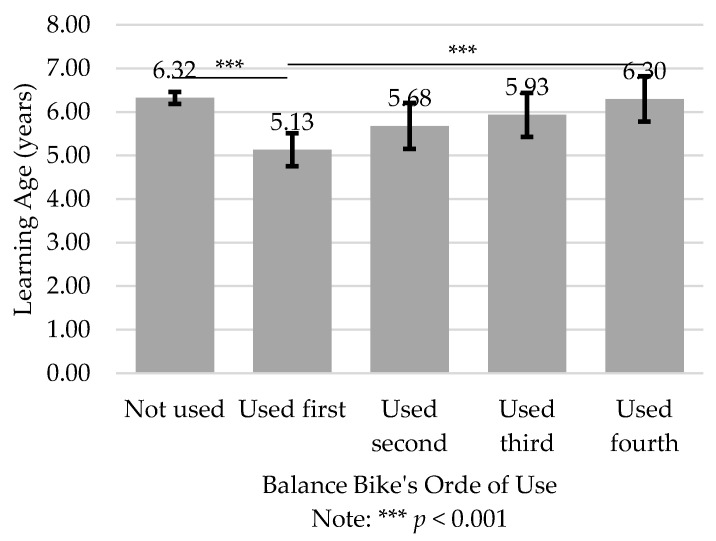
Learning age according to the order of use of balance bike; mean and 95% confidence interval.

**Table 1 ijerph-19-01814-t001:** Descriptive data regarding age and sex of the participants by decade and total.

Decades	Decimal Age (yrs)			Gender (n)			
	Mean ± Standard Deviation	Minimum	Maximum	Male	Female	Don’t Want to Say	Total
1960–69	55.05 ± 2.70	50.05	60.18	31	129	0	160
1970–1979	44.53 ± 2.75	39.91	50.29	119	238	2	359
1980–1989	35.65 ± 2.92	29.98	40.23	92	227	0	319
1990–1999	23.79 ± 2.88	19.92	30.23	209	236	1	446
2000–09	15.92 ± 3.20	10.13	20.21	251	214	3	468
2010–19	7.34 ± 1.81	2.39	10.35	142	109	2	253
Total	27.97 ± 14.7	2.39	60.18	844	1153	8	2005

**Table 2 ijerph-19-01814-t002:** Descriptive statistics of learning age according learning paths.

Learning Path	Participants	Mean ± Standard Deviation	95% Confidence Interval		Games Howell Significant Differences
			Lower	Upper	
1002	54	4.16 ± 1.34	3.80	4.53	All *** except 1203
1203	53	4.63 ± 1.44	4.23	5.03	All *** except 1002
1234	44	5.64 ± 1.99	5.03	6.24	0001 ***, 0123 ***, 1002 ***, 1203 ***
3124	42	5.90 ± 1.69	5.38	6.43	0001 ***, 1002 ***, 1203 ***
0102	630	5.97 ± 2.16	5.80	6.14	0001 ***, 1002 ***, 1203 ***
0123	364	6.03 ± 1.73	5.85	6.21	0001 ***, 1002 ***, 1203 ***, 1234 ***
0101	37	6.78 ± 2.98	5.79	7.78	0001 ***, 1002 ***, 1203 ***
0001	404	7.27 ± 3.74	6.90	7.63	All ***

Notes: first digit in learning path—balance bike; second digit—bicycle with two training wheels; third digit—bicycle with one training wheel; fourth digit—traditional bicycle; *** *p* ≤ 0.001.

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
