# Peer review of "Learning to Cycle: From Training Wheels to Balance Bike"

_ijerph, 2022, doi:10.3390/ijerph19031814_

Round 1
Reviewer 1 Report
The article is very interesting both for the topic it addresses and for the method adopted. In its originality, as well as its strengths, there are also some important weaknesses that must be overcome.
In particular, we ask you to:
- Justify with adequate bibliographic citations the retrospective method of collecting the learning age of the ability to ride a bicycle independently (LA);
- Analyse the presence or absence of differences in LA between subjects who have adopted the same learning paths in different decades;
In detail, the revision requests are as follows:
Introduction
Line 34 – substitute with “…this…”
Line 88 – substitute with “…The process…”
Line 93 – substitute with “…varies according to…”
Lines 99-102 - The sentence, perhaps, could be better written.
Line 124 – It would be useful to indicate the indicative date when the balance bike became available to Portuguese citizens.
Line 140 – substitute with “…to…”
Materials and methods
Line 171 - It would have been useful to ask parent instruction's level. On the basis of which bibliographic sources, or experimental data, can the correctness of the age of acquisition of the ability to ride a bicycle provided by older people be affirmed?
Table n.1 - The number of males and females and the total number of subjects in the decades do not show equivalent distributions. The (estimated) age of males is much lower than that of females. These elements must be indicated as possible limiting factors of the results obtained by the research.
Lines 208-209 – I probably do not have sufficient statistical skills to correctly understand the description of the “Sample calculation”. In any case, I ask you to explain the maximum number of groups considered (8) which, as far as I can understand, should instead be 12.
Results
It would be extremely useful (I suggest that it is necessary!) to compare the learning ages of autonomous cycling (LA) between subjects who adopt the same learning paths, particularly 1002 and 0001, in different decades.
Line 263 –To eliminate "other".
Discussion
Line 311 – A further element of weakness is the imprecision with which the age of learning of autonomous cycling (LA) could have been identified by the less young subjects of the sample.
Lines 311-312 – The sentence would be acceptable if LA values had occurred between subjects that adopted the same learning path in different decades.
Lines 320-321 – The sentence would be acceptable if you had also considered the time dimension. It is necessary to verify whether in younger subjects there is the same prevalence of the traditional approach as in older ones.
Line 351 – It is preferable to use the term "correlated to" rather than "influenced" because there can be many confounding factors.
Line 355 – Check the abbreviation: I think it should be (1002)
Lines 358-359 – The sentence needs to be rephrased by making it more dubious. Indeed, a cross-sectional survey with a population not well distributed by age group and with the collection of learning dates of the ability to ride a bicycle (LA) performed, by heart, many decades later, can have many causes of confusion of the results. Furthermore, It would be necessary to compare the learning ages of autonomous cycling (LA) between subjects who adopt the same learning paths, in this case, 1002 and 0102 in different decades.
Lines 365-366 – I suggest making the sentence doubtful for the same reasons indicated in lines 358-359.
Line 380 – Correct with “It…”
Line 390 – Change with "...at..."
Line 391 –Change with "... according to the ..."
Lines 404-405 –The sentence is not useful in this part of the article, better to remove it.
Line 417 –I suggest integrating with: "...among different cultures and genders can be ..."
Conclusions
I recommend reviewing the conclusions, possibly adopting a more dubious form, after verifying any differences in LA between subjects who have adopted the same learning paths.

Author Response
Dear Reviewer,
We would like to thank you for all your comments and suggestions. Your review was truly constructive and helped us to significantly deepen and improve our article.
We've taken all your comments and suggestions into account and posted our answers in the attached file. The only suggestion we could not address was to verify the existence of differences in the learning ages (LA) of subjects with the same learning path over the decades, due to the reduced size of some groups per decade.
Despite this aspect, we consider that the manuscript presents original data on a very relevant and increasingly popular topic, the learning to cycle. To the best of our knowledge, there is no other article addressing the topic of LA, specifically according to the learning paths and instruments. In this way, we believe that this information will attract the attention of readers for its originality and relevance.
Thank you very much on behalf of the entire team.
Best Regards.

Reviewer 2 Report
The present manuscript describes a study investigating the role of using the BB in the learning process of cycling independently. For this purpose, the authors created a questionnaire aimed at different age groups, divided into decades of birth. The main results are that: 1) BB usage has increased mainly since 2000-2009; 2) the use of the BB as the first cycling strategy is associated with a lower learning age.
However, a fundamental variable is missing from these results: the age of the first approach to any form of bicycle. If the BB is received very early, as it is likely to be in recent years, the first approach to cycling should be earlier than in previous decades. Without this fundamental information, the risk of BIAS is too high and the results cannot be taken into consideration.
Major revision
Thus, a second data collection including questions regarding this information is needed. If the authors can not retrieve the participants, due to the anonymity, they should repeat the data collection.
Minor revision
- figures. Significant differences must be shown into the figures.
- Figure 4. A legend for the learning strategies is needed as in the table.
- The questionnaire must be available for revision as supplementary material or as an appendix.
- Check the abbreviations, please. Learning age is often written in its expanded form.
- Figure 3 shows Balance bike’s percentage of use according to decades. What about other strategies according to decades? Maybe they should be mentioned in the text or a figure with multiple panels should be considered.
- Figure 4 shows Learning age according to learnings paths. How does the learning path change according to decades?
- Is the reference to private companies at line 296 necessary in this context?
- line 312-314: "Considering that the BB’s 312 use increased significantly in the last two decades, it is possible that the decrease in learning age is associated with the increase in the use of BB." This statement is unfounded if the age of the first approach to the bicycle is missing. Is it possible that the age of the first approach decreased in the last decades, together with the BB usage? Unfortunately, we can not answer this question.
- The weakness of the study must be described in deep. First, the retrospective nature of the study is a limit that must be commented. Secondly, the inclusion of different age groups may affect the quality of the answers: the recall time is longer for older participants and can affect the results. Younger and older participants may interpret questions differently.
Author Response
Dear Reviewer,
We would like to thank you for all your comments and suggestions. Your review was truly constructive and helped us to significantly deepen and improve our article.
Despite the weakness you indicated in your review, we consider that the manuscript presents original data on a very relevant and increasingly popular topic, the learning to cycle. To the best of our knowledge, there is no other article addressing the topic of LA specifically according to the learning paths and instruments. In this way, we believe that this information will fill a research caveat and attract the attention of readers for its originality and relevance.
Thank you very much on behalf of the entire team.
Best Regards.

Round 2
Reviewer 1 Report
I approve of the proposed revisions and only suggest some corrections.
Introduction
Line n. 100 – Delete the comma.
Results
Line 294 – I believe that only the mean values are shown in the graph.
Discussion
Lin n.338 – “genders”
Author Response
Introduction
Line n. 100 – Delete the comma.
Response: Deleted.
Results
Line 294 – I believe that only the mean values are shown in the graph.
Response: Thank you very much for the comment, in fact the legend was incorrect, we have already changed it.
Discussion
Lin n.338 – “genders”
Response: We changed genders to gender, but we believe the line mentioned by the reviewer was not correct.
Response:
Dear Reviewer,
On behalf of the entire team, we want to thank you for your review. We believe that it contributed for the improvement of the article.

Reviewer 2 Report
Dear Authors,
The first round of reviews significantly improved the manuscript quality. In particular, I can say that I am satisfied with how you have dealt with the minor revisions.
But unfortunately, I believe that the missing information I have asked you to provide is crucial. Without them, it is not possible to determine whether independent cycling is due to BB itself or to the fact that it is approached earlier than in the past.
This is a methodological error that invalidates the study results and, therefore, I do not think it can be accepted in the present form, although the field is exciting.
By this, I do not mean to devalue your work, which I believe is deserving and should continue in this direction. On the contrary, I am sure you will chart an important field of study with the necessary adjustments.
Author Response
Dear Reviewer,
We are glad to see that you considered that the manuscript has improved its quality with the 1st round. This improvement was due to feedback from both reviewers, so we would like to thank you for all your comments and suggestions.
We regret that you consider the missing information to be a methodological error. Although the BB can be used earlier by children, the BTW has been part of the culture of learning to ride a bike for longer. Therefore, this bike may end up being more affordable and used earlier than the BB. Regardless of the age at which children have their first contact with the bicycle, these data demonstrate that if children start with the BB (either by having the BB earlier, or by the facilitated learning with this bicycle), they end up acquiring cycling significantly earlier than children who started with the BTW. This information is new, and it can give rise to a new field of discussion and knowledge, with practical applications for an earlier cycle onset.
Once again, thank you very much for your contributions, they allowed us to improve the manuscript and deepen our reflection on the data.
